# Application of Synchronized Inertial Measurement Units and Contact Grids in Running Technique Analysis: Reliability and Sensitivity Study †

Đorđe Brašanac, Marko Kapeleti *, Igor Zlatović, Miloš Ubović and Vladimir Mrdaković

Faculty of Sport and Physical Education, University of Belgrade, Blagoja Parovića 156, 11030 Belgrade, Serbia; brasanacps@gmail.com (Đ.B.); igor.zlatovic@fsfv.bg.ac.rs (I.Z.); ubovicmilos@yahoo.com (M.U.); vladimir.mrdakovic@fsfv.bg.ac.rs (V.M.)

* Correspondence: marko.kapeleti@fsfv.bg.ac.rs

† This paper is an extended version of our conference abstract: Brašanac, Đ.; Kapeleti, M.; Zlatović, I.; Ubović, M.; Mrdaković, V. (2025). Kinematic and kinetic analysis of the Groucho running technique using inertial measurement units and contact detection devices. In Book of Abstracts from 22nd International Scientific Conference on Physical Activity and Health—Pledge for Life (p. 67). Faculty of Sport and Physical Education, Belgrade, Serbia, 6–7 December 2024; Dabović, M., Bubanja, I., Miletić, V., Eds. The link of the conference: https://www.fsfvconference.rs/ (accessed on 18 September 2025).

## Abstract

Background: Previous research has identified center of mass vertical oscillation and leg stiffness as the most common variables differentiating Natural and Groucho running techniques. The aim was to assess the inter-session reliability and inter-technique sensitivity of synchronized inertial measurement units and contact grids in quantifying kinematic and kinetic differences between Natural and Groucho running techniques. Methods: Eleven physically active and healthy males ran at a speed 50% higher than transition speed. Two sessions for Natural and two for Groucho running were performed, each lasting 1 min. Results: Most variables exhibited a similar inter-session reliability across running techniques, except contact time and center of mass vertical displacement, ranging from moderate to good (ICC = 0.538–0.897). A statistically significant difference between running techniques was found for all variables ($p < 0.05$), except for contact time and center of mass vertical oscillation ($p > 0.05$), likely due to inconsistency in reliability depending on the running technique, which may have covered the underlying differences. Conclusions: We can conclude that the combination of synchronized inertial measurement units and contact grids showed potentially acceptable reliability and sufficient sensitivity to recognize and differentiate between Natural and Groucho running techniques. The results may contribute to a broader understanding of the differences between these two running techniques and encourage the increased use of these devices within therapeutic, recreational, and sports running contexts.

**Keywords:** natural running; Groucho running; center of mass vertical oscillation; leg stiffness; kinematics; kinetics

## 1. Introduction

Inertial measurement units (IMUs) are multicomponent sensor systems designed to measure acceleration, angular velocity, and the orientation of a physical body in space. These systems consist of three key sensors—an accelerometer, a gyroscope, and a magnetometer—that collectively enable the estimation of kinematic parameters. The

accelerometer measures linear acceleration along three mutually perpendicular axes by detecting the displacement of an inertial mass relative to a reference frame. The gyroscope measures the angular velocity around three axes, whereas the magnetometer detects the strength and direction of the Earth's magnetic field, allowing for the determination of the device's absolute orientation in space. The reliability of IMUs has been confirmed in numerous studies examining different types of movements, such as walking, jumping [1], running [2] and various weightlifting exercises—Olympic lifts, squats, etc. [1,3]. By enabling detailed tracking of movement and contact phases, the use of IMUs helps identify movement patterns and optimize techniques to improve performance and reduce the risk of injury [4–7].

Contact grids (CGs) consist of two units, which together create a "carpet" of infrared light a few millimeters above the floor. When the light beams are interrupted, the time is recorded, thus allowing detection of contact and flight times. These devices have a wide range of applications in biomechanical analysis, such as running and jumping [8–10]. The IMU sensor can be combined with other measuring devices—such as, for example, CGs—in order to achieve more precise and comprehensive motion analysis. If synchronized within the common software, they can offer broader spectrum of already predefined and precalculated variables, facilitating the comprehensiveness and time efficiency of analysis when using these devices in practical settings. Thus, the combination of IMUs and CGs could further expand the scope of biomechanical movement analysis, offering broader applications and deeper insights.

Running is a fundamental form of human locomotion that is often analyzed from a biomechanical perspective to better understand its basic principles. Research in this field has focused on the forces acting on the body, movement patterns, and factors influencing performance [11–14]. Vertical oscillation of the body's center of mass is one of the most commonly used reference points in the biomechanical analysis of running [15–17]. Another important running factor is the stiffness of the locomotor system, defined as the extent to which an object resists deformation in response to an applied force [18], which reflects the ability to accumulate, store, and release energy from elastic deformation [19]. All of this could be analyzed with a combination of IMUs and CGs by utilizing their technological ability to estimate displacements and ground reaction forces during movement on the basis of acceleration and contact detection. Since acceleration is the second derivative of displacement, integrating the acceleration signal twice (with proper filtering) can yield estimates of velocity and displacement. On the other hand, contact time, measured using CGs, serves as a basis for assessing oscillatory or spring-like behavior of the body, particularly during running.

An optimal level of leg stiffness can enhance running performance [20] and reduce injury occurrence [21]. Runners frequently use various strategies to optimize stiffness, including modifying surfaces, footwear, and running techniques [22,23], such as the Groucho running technique [20]. This technique is characterized by reduced vertical oscillation of the center of mass, lower leg stiffness, and increased hip and knee flexion [24], thus reducing loading on the musculoskeletal system and increasing muscle activation and oxygen consumption [25,26], potentially helping to prevent injuries while enhancing the effectiveness of training within a specific timeframe.

Research on the biomechanical differences between the Natural and Groucho running techniques has shown that the Groucho technique is associated with a longer contact time, shorter flight phase [27,28], shorter stride length [24], higher step frequency [28], reduced vertical oscillation of the center of mass [24], and lower leg stiffness [24]. Most previous studies have analyzed and compared these two running techniques via motion capture systems [27] or a combination of kinematic analysis and force platforms [29]. Despite

the widespread use of IMU and CG devices in the biomechanical analysis of running techniques, the analysis and comparison of Natural and Groucho running techniques has not yet been examined using these devices.

The aim of this study is to assess the inter-session reliability and inter-technique sensitivity of IMUs and CGs in quantifying previously established differences between Natural and Groucho running techniques. Given the technological characteristics and physics principles underlying these devices, it is hypothesized that the measurements will demonstrate an acceptable level of reliability and that they will be sufficiently sensitive to detect differences in kinematic and kinetic characteristics between these two techniques. The results may contribute to a broader understanding of the differences between these two running techniques and encourage the increased use of these devices within therapeutic, recreational, and sports running contexts by facilitating more comprehensive and time efficient analysis when used in practical settings.

## 2. Materials and Methods

### 2.1. Sample of Subjects

The study included 11 physically active and healthy male participants without recent injuries to the locomotor system and previously inexperienced in Groucho running. The participants' age was $23.1 \pm 2.5$ years, body height was $182.7 \pm 5.8$ cm, body mass was $83.1 \pm 7.4$ kg, and body mass index was $24.9 \pm 1.6$ kg/m$^2$. Each subject signed a written informed consent form confirming their voluntary participation in the experiment. Achieved statistical power for this sample size was calculated using G*Power software v3.1.9.7 (Heinrich Heine University, Düsseldorf, Germany). Analysis for two-tailed dependent (paired) T-test showed that 11 participants produced moderate 67% statistical power in detecting large effect size f (0.4), at an alpha level of 0.05.

### 2.2. Experimental Protocol

The protocol consisted of basic anthropometric and morphological measurements (body height, body mass, and dominant leg length), followed by a ten-minute warm-up through cyclic exercises, stretching, and muscle activation drills. The next step involved determining the transition speed from walking to running, followed by familiarization with the Groucho running technique, and finally, the main part of the experimental protocol—variable sampling during Natural and Groucho running techniques. The subjects were instructed not to consume food or liquids for at least one hour before the testing session and not to engage in any intense physical activity for at least 24 h before the experiment. The experiments reported in the article were undertaken in compliance with the relevant laws and institutional guidelines. The experimental protocol was planned and conducted in accordance with the Declaration of Helsinki and was approved by the Ethics Committee of the Faculty of Sport and Physical Education, University of Belgrade (Date: 9 April 2025; Number: 02-506/25-2).

Body height was measured via a digital altimeter (BSM170, Arab Engineers). The measurement procedure was conducted following the manufacturer's recommendations. Body mass index was calculated as the ratio between body mass [kg] and body height [m] squared. Leg length measurement was required for the calculation of leg stiffness via software (see the Variables section).

Running speed was standardized across participants based on transition speed from walking to running in order to minimize the differences in Natural running technique between the subjects, as transition speed accounts for variations in anthropometric characteristics and motor abilities important for running at the particular speed [30]. The transition speed determination protocol involved testing under natural walking conditions

on a treadmill while continuously holding the handles. The participants were instructed that the goal was not to determine their maximum walking speed but rather the speed at which walking or running felt more natural and easier. The treadmill display showing speed was hidden, and participants were free to choose the more suitable movement pattern for a given speed. The initial walking speed was 5 km/h, and the speed was gradually increased every 30 s by either 0.4 km/h or 0.2 km/h depending on the proximity to the transition speed. The objective indicator of transition speed was the presence of a flight phase, whereas the subjective indicator was the participant's feedback. After the initial transition speed determination, a verification test was performed at the given speed following a 30 s break. In this sample of subjects, the transition speed was $7.1 \pm 0.4$ km/h.

Familiarization with the Groucho running technique was conducted through verbal instructions, demonstrations, participant practice, and corrections made by the examiner. The criteria for proper Groucho running technique included visibly reduced vertical oscillation of the center of mass, continuous maintenance of the center of mass at a constant level, and increased flexion at the hip and knee joints of both legs. The exact verbal instructions were as follows: "Flex your knees, lower your body height while running, keep the height at a constant level and reduce the impact of the foot". Familiarization lasted around 5–10 min, depending on the subject learning efficiency, until the subject could perform proper Groucho technique continuously and independently. Familiarization for Natural running technique was unnecessary because it did not involve any verbal cues in order to simulate real natural conditions for an individual.

The participants ran at a supratransitional speed—their individual transition speed increased by 50%, to reduce the variability of relatively slow running on transition speed, as it is known that increases in running speed reduces the coordination variability during stance phase [31]. For this sample of subjects, the running speed was $10.6 \pm 0.5$ km/h. Two running sessions were performed per technique in the following order: (1) Natural running and (2) Groucho running, each lasting 1 min, with a 2 min break between techniques and a 10 min break between sessions. Techniques were not randomized within a session across participants because Natural running was performed without verbal instructions, so it did not affect performance during Groucho running as there is no learning and transfer effect, as well as fatigue occurrence due to controlled running intensity and the duration of resting period. Technique control during the experiment was conducted through visual observation, with corrections provided via previously explained verbal instructions from the examiner.

### 2.3. Testing Devices

Two complementary measuring devices (IMU and CG) made by the Norwegian company MuscleLab (Figure 1) were used. The IMU consisted of an accelerometer (16G, accuracy $\pm 1\%$), a gyroscope (2000 dps, accuracy $\pm 1\%$), and a magnetometer (1300/2500 µT, accuracy $\pm 5\%$). The sampling frequency of the IMU was 200 Hz. The resolution of the CG was < 2 ms. Synchronization was achieved via a data synchronization unit (accuracy $\pm 1$ ms) with a built-in radio host device (2.4 GHz), whereas the software enabled data acquisition and processing (v10.241.11.4.5359, x64 bit). The IMU sensor was placed at the L5–S1 level to approximate the center of mass location, whereas the CGs were placed in line with the treadmill and separated 3 m apart.

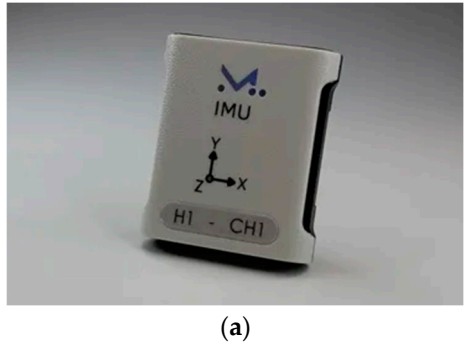
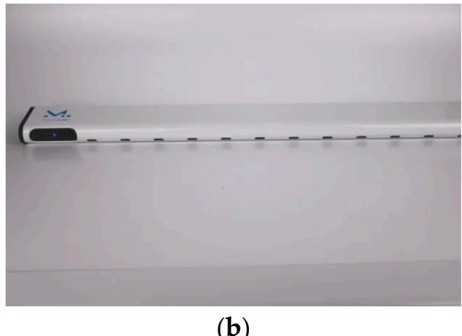

<div align="center">(<b>a</b>)          (<b>b</b>)</div>

**Figure 1.** Inertial measurement unit (**a**) and contact grid (**b**) made by the Norwegian company MuscleLab https://www.musclelabsystem.com/products/ (accessed on 1 October 2025).

### 2.4. Variables

The following variables were acquired:
(1) Contact time—CT [ms]
(2) Flight time—FT [ms]
(3) Step length—SL [cm]
(4) Step frequency—SF [step/s]
(5) Center of mass takeoff angle—$COM_{angle}$ [°]
(6) Center of mass vertical oscillation—$O_{vert}$ [cm]
(7) Leg stiffness—$K_{leg}$ [kN/m]

CT, FT, SL and SF are the most common kinematic variables used for running analysis [32,33]. The $COM_{angle}$ refers to the angle between the horizontal axis and the resultant velocity vector of the COM at the moment the subject leaves the ground. We utilized this variable in order to describe the differences between techniques in a more comprehensive manner, given that different takeoff angle under the same impulse influence the achieved height of the object in flight, and consequently $O_{vert}$. Software computed $K_{leg}$ on the basis of the spring-mass model, utilizing leg length and joint kinematics. Specifically, leg compression ($\Delta L$) was estimated by calculating changes in effective leg length during the stance phase on the basis of initial leg length, running velocity and contact time [34] which were all derived from predefined anthropometric data and velocity, as well as IMU-measured body motion and CG detection. The peak vertical force ($F_{max}$) was approximated via the vertical displacement recorded by the IMU and determined via double integration of the vertical acceleration over time [35]. $K_{leg}$ was then computed as the ratio of $F_{max}$ to $\Delta L$ [34].

### 2.5. Statistical Analysis

Each variable was represented by its mean value and standard deviation. Data were subjected to normality distribution analysis via the Shapiro–Wilk test (for both sessions and techniques). Reliability analysis was conducted by comparing both sessions within technique via the average measures intraclass correlation coefficient (ICC) with a two-way mixed-effects model and absolute agreement, presented with a 95% confidence interval (CI) and standard error of measurement (SEM). The following criteria were used for ICC classification: poor < 0.50, moderate 0.50–0.74, good 0.75–0.89, and excellent ≥ 0.90 [36]. Systematic errors within technique were examined through repeated-measures analysis of variance. Sensitivity analysis was conducted by comparing the second sessions of both techniques via dependent T-test. Effect size (ES) for the differences between running techniques was calculated via Cohen's D as the difference between the mean values divided by the pooled standard deviation. The effect size was considered small if it was ≤0.49, medium if it was between 0.50 and 0.79, and large if it was ≥0.80 [37]. Statistical data

processing was performed via IBM SPSS Statistics v20.0.0 (IBM Corp., Armonk, NY, USA). The level of statistical significance was set at $\leq 0.05$.

## 3. Results

The Shapiro–Wilk test confirmed a normal distribution of data for all variables in both sessions and for both techniques in this sample of subjects ($p > 0.05$). The results of the repeated-measures analysis of variance within technique did not show significance, indicating that no systematic error was detected ($p > 0.05$). Figure 2 shows the representative plots of $O_{vert}$ and $K_{leg}$ during 10 s across running techniques. Figures 3 and 4 show the descriptive statistics for all the variables across running techniques. Tables 1 and 2 present the results of the inter-session reliability analysis for the Natural and Groucho running techniques, respectively. Table 3 presents the results of the inter-technique sensitivity analysis.

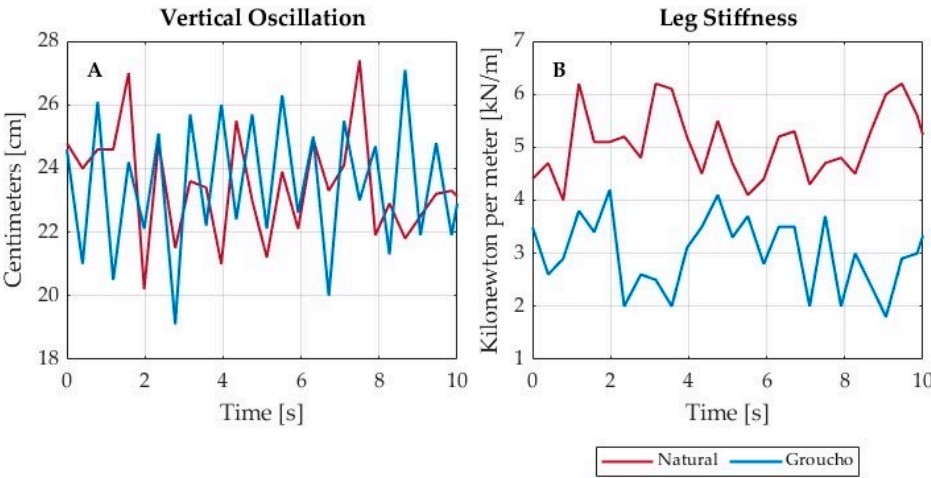

**Figure 2.** Representative plot for (**A**) vertical oscillation and (**B**) leg stiffness during Natural and Groucho running.

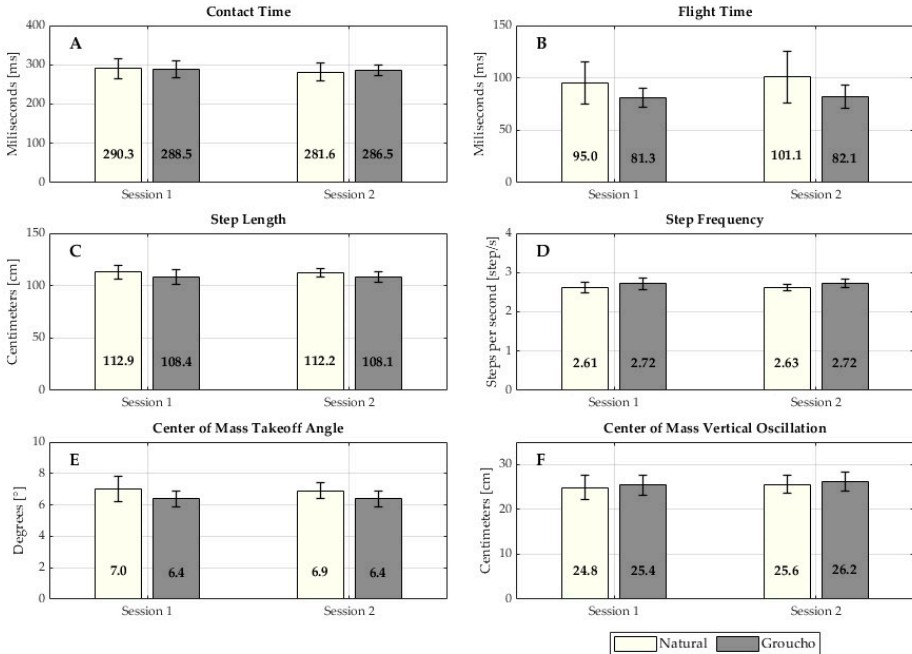

**Figure 3.** Means ± standard deviations for all kinematic variables during Natural and Groucho running—(**A**) contact time, (**B**) flight time, (**C**) step length, (**D**) step frequency, (**E**) center of mass takeoff angle and (**F**) center of mass vertical oscillation.

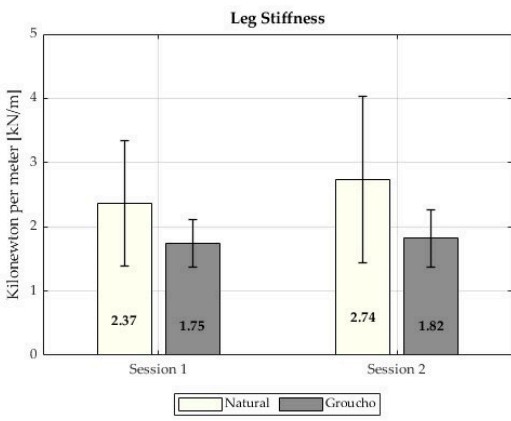

**Figure 4.** Means ± standard deviations for leg stiffness during Natural and Groucho running.

**Table 1.** Results of the ICC and SEM analysis for the Natural running technique.

| Variables | ICC | 95% CI | SEM | $p$ |
|---|---|---|---|---|
| CT | 0.575 | −0.487–0.884 | 17.9 | 0.095 |
| FT | 0.690 | −0.114–0.916 | 17.5 | 0.040 * |
| SL | 0.897 | 0.626–0.972 | 2.5 | 0.001 * |
| SF | 0.845 | 0.433–0.958 | 0.07 | 0.004 * |
| $COM_{angle}$ | 0.884 | 0.566–0.969 | 0.3 | 0.001 * |
| $O_{vert}$ | 0.684 | −0.084–0.913 | 1.3 | 0.039 * |
| $K_{leg}$ | 0.720 | 0.030–0.923 | 0.88 | 0.027 * |

CT—contact time; FT—flight time; SL—step length; SF—step frequency; $COM_{angle}$—center of mass takeoff angle; $O_{vert}$—COM vertical oscillation; $K_{leg}$—leg stiffness. * Statistical significance ($p \leq 0.05$).

**Table 2.** Results of the ICC and SEM analysis for the Groucho running technique.

| Variables | ICC | 95% CI | SEM | $p$ |
|---|---|---|---|---|
| CT | 0.843 | 0.410–0.958 | 12.9 | 0.005 * |
| FT | 0.622 | −0.546–0.901 | 17.4 | 0.082 |
| SL | 0.883 | 0.555–0.969 | 1.8 | 0.002 * |
| SF | 0.800 | 0.216–0.947 | 0.04 | 0.012 * |
| $COM_{angle}$ | 0.822 | 0.340–0.952 | 0.4 | 0.007 * |
| $O_{vert}$ | 0.868 | 0.516–0.964 | 0.9 | 0.001 * |
| $K_{leg}$ | 0.538 | −0.858–0.878 | 0.88 | 0.131 |

CT—contact time; FT—flight time; SL—step length; SF—step frequency; $COM_{angle}$—center of mass takeoff angle; $O_{vert}$—COM vertical oscillation; $K_{leg}$—leg stiffness. * Statistical significance ($p \leq 0.05$).

**Table 3.** Results of dependent T-test and effect size analysis between running techniques.

| Variables | t | $p$ | Cohen's D |
|---|---|---|---|
| CT | −0.802 | 0.441 | 0.26 |
| FT | +2.663 | 0.024 * | 1.00 |
| SL | +5.747 | 0.000 * | 0.90 |
| SF | −5.078 | 0.000 * | 0.94 |
| $COM_{angle}$ | +4.627 | 0.001 * | 1.00 |
| $O_{vert}$ | −1.464 | 0.174 | 0.30 |
| $K_{leg}$ | +2.469 | 0.033 * | 0.95 |

CT—contact time; FT—flight time; SL—step length; SF—step frequency; $COM_{angle}$—center of mass takeoff angle; $O_{vert}$—COM vertical oscillation; $K_{leg}$—leg stiffness. * Statistical significance ($p \leq 0.05$).

For almost all the variables, the ICC was statistically significant, except for CT, which tended toward statistical significance. The ICCs ranged from moderate to good for all the variables (0.575–0.897). In 3 out of 7 cases, inter-session reliability was good. Moderate

reliability was observed for CT, FT, $O_{vert}$, and $K_{leg}$, whereas good reliability was observed for SL, SF, and $COM_{angle}$.

For almost all the variables, the ICCs were statistically significant. $K_{leg}$ was not statistically significant, whereas FT tended toward statistical significance. The ICCs ranged from moderate to good for all the variables (0.538–0.883). In 5 out of 7 cases, inter-session reliability was good. Moderate reliability was observed for FT and $K_{leg}$, whereas good reliability was observed for CT, SL, SF, $COM_{angle}$, and $O_{vert}$.

A statistically significant difference between running techniques was found for all the variables, except for CT and $O_{vert}$. Figures 3 and 4 show that FT, SL, $COM_{angle}$, and $K_{leg}$ are lower, whereas SF is higher in the Groucho running technique.

## 4. Discussion

The aim of this study was to examine the inter-session reliability and inter-technique sensitivity of IMUs and CGs in detecting differences in kinematic and kinetic characteristics between Natural and Groucho running techniques. The devices demonstrated, in most cases, nearly good-to-good reliability in measurements and are sufficiently sensitive to detect differences in most of the monitored variables, which could be considered adequate in the measurements of biological systems. The significance of this study lies in providing new insights into the application of synchronized IMUs and CGs in both scientific research and practical fields related to running, as this configuration may enhance capability and efficiency of running analysis.

Most variables exhibited a statistically significant ICC, with inter-session reliability ranging from moderate to good (ICC = 0.538–0.897) (Tables 1 and 2). Specifically, moderate reliability in both techniques is demonstrated for FT and $K_{leg}$, good reliability is demonstrated in both techniques for SL, SF and $COM_{angle}$, and the reliability of CT and $O_{vert}$ differs from moderate to good depending on the running technique. No variables reached poor level of reliability. As previously mentioned, the adequate reliability of these devices has been confirmed in numerous studies examining different movement patterns, such as walking and jumping [1], running [2], and various weightlifting movements, including Olympic lifts and squats [1,3].

Previous research on this topic has identified $O_{vert}$ and $K_{leg}$ as the most common variables differentiating these two running techniques [24,38]. The present results reveal a statistically significant difference between the running techniques for most analyzed variables, except for CT and $O_{vert}$ (Table 3), probably because of inconsistency in reliability depending on the running technique, and relatively lower reliability in the Natural compared to Groucho running technique (Tables 1 and 2). ESs of all variables are in accordance with the statistical significance, that is, variables that showed significant differences between techniques had large ES (Table 3). The lower FT, SL, $COM_{angle}$, and $K_{leg}$ values, along with an increased SF in the Groucho technique, suggest that this technique involves shorter, more frequent, and "softer" steps, which aligns with previous research findings [24,27,28].

Previous studies have reported that the Groucho running technique results in longer CT and lower $O_{vert}$ values [24,27,28,38]. A partial explanation for the lack of observed differences in this study is the inconsistency in the reliability of CT and $O_{vert}$ across running techniques (predominantly in Natural), which may have covered the underlying differences. Regarding $O_{vert}$, this finding is not in line with previous studies that confirmed the reliability of this variable [39], while CT has the tendency to show relatively lower validity and reliability via IMUs [40,41]. Additionally, it is hypothesized that the reduced $K_{leg}$ in the Groucho technique alters vertical oscillation—being more pronounced in the eccentric phase and less pronounced in the concentric phase than in the Natural running technique—which may partially explain this unexpected result. In support of this hypothe-

sis, the lower $COM_{angle}$ observed in the Groucho technique likely reduces the height of the COM trajectory, leading to less upward vertical oscillation after takeoff. Furthermore, it is unlikely that this unexpected result stems from methodological issues, as participants had sufficient familiarization that enabled them to include every technical request in the Groucho running described in the exact verbal instructions, data from the second session were used for sensitivity analysis, the IMU was positioned on an appropriate location and equipment was well synchronized. The authors of this study do not necessarily imply that CT and $O_{vert}$ are not suitable for distinguishing the differences between Natural and Groucho running techniques and running technique analysis in general, but that these variables should be re-examined on reliability in different research settings along with joint kinematic analysis to describe the movement more comprehensively.

Since the only way to validate the accuracy of these devices is to directly compare the differences in biomechanical variables between these running techniques, with findings from previous studies that employed more direct measurement methods and given the previously stated explanations for CT and $O_{vert}$, we can conclude that the combination of synchronized IMUs and CGs showed potentially acceptable reliability and sufficient sensitivity to recognize and differentiate between Natural and Groucho running techniques. This study contributes to a better understanding of the capabilities and potential applications of synchronized IMUs and CGs for biomechanical running analysis, as this technological configuration has the potential to analyze efficiently more variables than using, for example, only CGs for running technique analysis. The primary uncertainty pertains to the CT and $O_{vert}$ variables, which requires further investigation. One of the study limitations is the relatively smaller sample size, leading to moderate statistical power (67%) and lower chance of detecting true differences (risk of type II error), due to which we cannot confidently conclude if the differences in CT and $O_{vert}$ between running techniques ultimately exist or not. Reflecting on the limitations of this study, future research should include a larger sample size and a population of professional runners already familiar with the Groucho running technique, in order to examine the generalizability of these findings. Also, it is suggested to explore potential differences among different device manufacturers and across different movements, in order to confirm the accuracy of these findings. Moreover, it is crucial to validate their use in this context by comparing them with the gold standard—motion capture systems.

## 5. Conclusions

All variables monitored through synchronized IMU and CG exhibited moderate to good inter-session reliability, while most of them were sensitive enough to differentiate between Natural and Groucho running techniques. According to these results, we can conclude that synchronized IMUs and CGs showed potentially acceptable reliability and sufficient sensitivity to recognize and differentiate between Natural and Groucho running techniques, which could be considered adequate in the measurement of biological systems. This study contributed to a better understanding of the potential of utilizing synchronized IMUs and CGs, thereby encouraging their broader application in running-related research and practice. Future studies should re-evaluate the measurement properties of synchronized IMUs and CGs and compare it with gold standard (motion capture system), while utilizing different device manufacturers on a larger sample size of professional runners already familiar with the Groucho running technique.

**Author Contributions:** Conceptualization, M.K. and V.M. Data curation, Đ.B. and M.K. Formal analysis, Đ.B., M.K. and I.Z. Investigation, Đ.B., M.K., I.Z. and M.U. Methodology, M.K., I.Z. and V.M. Project administration, Đ.B. and M.U. Resources, V.M. Supervision, V.M. Validation, Đ.B., M.K., I.Z. and M.U. Visualization, Đ.B. and M.U. Writing—original draft, Đ.B. and M.K. Writing—review

and editing, I.Z., M.U. and V.M. All authors have read and agreed to the published version of the manuscript.

**Funding:** No financial or material support of any kind was received for the work described in this article.

**Institutional Review Board Statement:** The experiments reported in the article were undertaken in compliance with the relevant laws and institutional guidelines. The experimental protocol was planned and conducted in accordance with the Declaration of Helsinki and was approved by the Ethics Committee of the Faculty of Sport and Physical Education, University of Belgrade (Date: 9 April 2025; Number: 02-506/25-2).

**Informed Consent Statement:** Each subject signed a written informed consent form confirming their voluntary participation in the experiment.

**Data Availability Statement:** The raw data supporting the conclusions of this article will be made available by the authors on request.

**Conflicts of Interest:** The authors declare no potential or actual conflict of interest that could inappropriately influence (bias) their work.

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
