# Peer review of "Application of Synchronized Inertial Measurement Units and Contact Grids in Running Technique Analysis: Reliability and Sensitivity Study"

_2673-7078, doi:10.3390/biomechanics5040079_

Round 1

Reviewer 1 Report

Comments and Suggestions for Authors

This study explored the reliability and sensitivity of synchronized inertial measurement units (IMUs) and contact grids (CGs) in distinguishing between natural and Groucho running techniques. The authors validated the repeatability and discriminatory validity of both devices from the perspective of multiple kinetic and kinematic variables. The study provides an experimental basis for a wearable approach to running technique analysis with some promising applications. However, there are still obvious problems with the current manuscript in terms of scientific rigor, experimental design, statistical analysis, and interpretation of results. Specific revision suggestions are listed below:

  1. The greatest contribution of the current research is vaguely defined. Please clearly emphasize why the technological or applied breakthroughs in this research are “necessary” and “irreplaceable” compared to existing IMU research.
  2. In addition, it is recommended to review the latest research related to IMU in biomechanical applications. To provide more effective evidence, the authors may consider referring to the following updated relevant studies: Data-Driven Deep Learning for Predicting Ligament Fatigue Failure Risk Mechanisms (https://doi.org/10.1016/j.ijmecsci.2025.110519).
  3. Inclusion of only 11 healthy male subjects with no professional running background limits extrapolation. Please add a discussion of this limitation or justify the sample size through efficacy analysis.
  4. Proficiency is an important factor influencing the fluctuation of the variable, but the text only replaces formal training with “verbal instruction and practice”, which lacks quantitative criteria. It is recommended that the indicators of assessment before inclusion be clarified or that subjective scores be added.
  5. Was the sequential arrangement of the two runs randomized? Was there a learning or fatigue effect? Please specify whether a randomized crossover design was used and how it was statistically controlled.
  6. Although the authors mention the potential of the IMU+CG combination, they do not compare it with the “gold standard”, such as optical motion capture systems. It is recommended that the need for this be added to the literature or emphasized in the discussion.
  7. Please add why these 7 variables (especially the COM angle) were chosen and their relationship to running performance or injury risk should be described in the introduction or methods.
  8. Figure 1 is only a photo of the device, and it is recommended to add comparison plots of key variables (e.g., Kleg, Overt) in the two running conditions to enhance the readability of the paper. Additionally, it is recommended that the results section be presented in graphical form.
  9. The current conclusion asserts that “two technologies can be identified”, and it is suggested that this be amended to read “showed some potential in a small sample of preliminary tests” to avoid exaggerating the results.

Reviewer 2 Report

Comments and Suggestions for Authors

The article entitled “Application of Synchronized Inertial Measurement Units and Contact Grids in Running Technique Analysis: Reliability and Sensitivity Study” aimed to assess the reliability and sensitivity of synchronized inertial measurement units (IMUs) and contact grids (CGs) in quantifying kinematic and kinetic differences between Natural and Groucho running techniques.

The authors state in the introduction that devices such as inertial measurement units and contact grids, when combined, can serve as effective instruments for biomechanical assessment in modalities like running, for example. However, the authors don't make it clear what type of race they are addressing. The kinetics and kinematics of sprint races are quite different from those of endurance races. The location of the race (road, track, or trail) will also influence the evaluation metrics proposed in this study. The authors must convey these points in this text.

Why did the authors choose the transition speed between walking and running for evaluation, since this is not one of the most common metrics for running training? Running economy tests are well-established for assessing running efficiency. Therefore, transition speed does not appear to be the most appropriate choice for such an assessment. Therefore, it is also important to make clear in the text which audience and type of race the authors are addressing in this article.

The authors also state that different training can be used to improve various parameters that can help improve running and including the Groucho running technique. However, there is very little scientific evidence on the benefits of applying this technique to improve running. A quick search of databases such as PUBMED, SCOPUS, and SPORTS DISCOS revealed only 11 articles addressing this technique as a potential improvement in running.

Depois disso, os autores buscam uma comparação entre a avaliação da técnica natural decorrida com o Groucho, porém, não explicam qual é a bases de fundamentação para caracterização da chamada corrida natural. Como esta é uma técnica muito pouco validade cientificamente, não é possível utilizar os valores de ICC para afirmar que as diferenças são relevantes nem que as estimativas propostas pela interação entre os devices utilizados neste artigo sejam adequadas.

The authors then seek to compare the natural technique evaluation with the Groucho technique, but they do not explain the basis for characterizing the so-called natural running. Because this technique lacks scientific validity, it is not possible to use the ICC values to assert that the differences are significant or that the estimates proposed by the interaction between the devices used in this article are adequate. Furthermore, as the race execution times are very different, it is expected that the values presented will also be different.

For statistical analysis, the authors claim to have used ANOVA to compare groups and time points. However, the authors used an N of 11 participants. Although homogeneity, sphericity, and normality were tested and ensured before applying ANOVA, the basic assumption for applying this statistical model is an N of at least 30 for each observation time point. More specific models, such as generalized mixed models, exist for comparisons across groups and time points with small N.

Minor Comments

Page 3, line 126, the authors state that BH was evaluated as well as BMI and BM. What are BH, BMI, and BM? There is no prior description of this acronym in the text.

Reviewer 3 Report

Comments and Suggestions for Authors

This study investigated the feasibility of distinguishing Natural and Groucho running techniques using metrics derived from data collected via an IMU positioned near the center of mass and contact grids adjacent to the treadmill. The findings indicate that several metrics may effectively differentiate between these running techniques. The manuscript is clearly written and well-organized, with thorough descriptions of the methodology and analysis. Further discussion and additional analyses could enhance the overall strength of the work.

Lines 107 and 143: Given that the goal of the study was to distinguish the two running techniques, was information collected on the participants’ prior experience with Natural and Groucho running techniques? For those who might not be as familiar with Groucho running, how might longer exposure to this technique influence the conclusion of the study?

Lines 143, 146 and 155: How long was the familiarization with the Groucho running technique? Were the participants instructed to run with reduced vertical oscillation of the center of mass, which was one of the estimated variables? Will the authors provide the exact text of the verbal instructions in the paper?

Line 152: Was the order of the techniques the same for all participants? If so, how might this ordering have affected the outcomes of the study?

Line 187: How did the authors decide which sessions were subjected to which type of analysis?

Line 235: It seems the highlight (in red) wasn’t saved correctly.

Lines 263 and 271: Do the authors suggest that vertical oscillation of the center of mass is not an ideal metric for distinguishing Natural from Groucho techniques? How does the reliability in this study compare with previous studies?

Lines 279 and 292: How was sufficient familiarization justified? Compared with the participants in this study, how do the authors expect the conclusions to hold for professional runners, who may represent a more suitable target population?

Round 2

Reviewer 1 Report

Comments and Suggestions for Authors

All comments have been addressed.

Author Response

Dear Reviewer, thank you very much for supporting our work!
Best regards from all authors!

Reviewer 3 Report

Comments and Suggestions for Authors

The authors have addressed my comments. I have no further feedback.

Author Response

(The authors gave the same response as above.)
